# Blood-Based miRNA Panels for Timely Detection of Non-Small-Cell Lung Cancer: From Biomarker Discovery to Clinical Translation

**DOI:** 10.3390/ijms262412035

**Published:** 2025-12-14

**Authors:** Yazan Zedan, Maria Yurievna Konoshenko, Olga Evgenievna Bryzgunova, Antonina Aleksandrovna Ilyushchenko, Yaroslava Mikhailovna Danilova, Stanislav Dmitrievich Gorbunkov, Kirill Alekseevich Zykov, Pavel Petrovich Laktionov

**Affiliations:** 1Institute of Chemical Biology and Fundamental Medicine Siberian Branch Russian Academy of Sciences ICBFM SB RAS, Academician Lavrentyev Avenue 8, Novosibirsk 630090, Russia; y.zedan@g.nsu.ru (Y.Z.); lacyjewelrymk@gmail.com (M.Y.K.); lakt@1bio.ru (P.P.L.); 2Faculty of Natural Sciences, Novosibirsk State University, Novosibirsk 630090, Russia; 3Research Institute of Pulmonology of the Federal Medical-Biological Agency of the Russian Federation, Orekhoviy Boulevard 28, Moscow 115682, Russia; yaroslava.danilova.82@mail.ru (Y.M.D.); secretary@pulmonology-fmba.ru (S.D.G.); kirillaz@inbox.ru (K.A.Z.)

**Keywords:** LC, NSCLC, miRNA, timely detection, early stages, miRNA panels

## Abstract

Lung cancer (LC) remains a leading cause of global cancer mortality, driving the need for novel timely detection strategies, i.e., stages I–II detection when tumor curation is efficient. Circulating microRNA (miRNAs), with their unique stability in biofluids, offer a powerful approach for non-invasive detection. This review compiles validated miRNAs implicated in the early stages of non-small-cell lung cancer (NSCLC), elucidating their roles in key oncogenic pathways such as epithelial-mesenchymal transition (EMT), *PI3K*/*AKT*/*mTOR*, and *JAK*-*STAT*, which regulate proliferation, apoptosis, and metastasis. Furthermore, we critically evaluate developed miRNA panels with a specific focus on advanced quantification and normalization strategies, including exogenous spike-in controls and data-driven methods like pairwise normalization, to enhance diagnostic accuracy. Consequently, we identify and rank the most viable miRNA candidates according to key analytical and clinical metrics, providing a clear roadmap for translating these biomarkers into effective panels for the timely detection of NSCLC.

## 1. Introduction

Since lung cancer (LC) is considered one of the most common and deadliest types of cancer, great attention is paid to its diagnosis and treatment [1]. The vast gas exchange surface of the lungs, while facilitating exposure to carcinogens like tobacco smoke, also renders early-stage lesions exceptionally difficult to detect with conventional methods [2,3]. The risk of lung cancer development is 20–40 times higher in lifelong smokers compared to non-smokers. Moreover, second-hand smoke exposure increases the risk of lung cancer by more than 20% among never-smokers [4]. Of course, heredity, somatic mutations and immune system failures contribute to the development of lung cancer, whether related to the contact of lung cells with chemical or physical carcinogens or not [5]. With all this, lung cancer usually develops asymptomatically, and 60–70% of patients are at advanced stage IV at the time of diagnosis [6]. The 5-year survival rate for LC patients is 80–85% at stage I [7], and 63.1% in pN0, 51.9% in pN1, and 33.5% in pN2 of stage II [8], but at the later stages it tends to be 16% [6], demonstrating an urgent need for early lung cancer screening. No less than 80% of all LC cases are non-small-cell LC (NSCLC, including adenocarcinoma (AD) and squamous cell carcinoma (SCC)), with the remaining 10–15% being small-cell lung cancer (according to the American Cancer Society [cancer.org |1.800.227.2345]). In this review, we will concentrate on early-stage NSCLC diagnostics using promising molecular markers from the circulating blood.

A number of methods are used for NSCLC diagnostics, including CT, MRI, PET, IA (immunoassay), IGH (immunohistochemistry), FISH, mass spectroscopy assay of volatile organic compounds (VOCs), and liquid biopsy methods [9,10]. Computed tomography (CT) has been shown to be a very convenient method for early NSCLC screening of smokers, and it demonstrates a 20% reduction in lung cancer mortality with annual CT screening [11]. Nevertheless, limited access to modern imaging equipment and qualified personnel, cost and time of the procedure, and high false-positive rates (up to almost 50%) [12] do not allow this procedure to be considered a screening test. MRI and PET have similar limitations (except for the specificity of PET). A separate problem is the spread of lung cancer among non-smokers. In Asia, for example, in recent years, 45% to 70% of all patients with lung cancer in this population group have been non-smokers [13]. Most of the listed methods used to diagnose NSCLC cannot be considered screening methods, and at present, only immunoassay of tumor markers, analysis of tumor-related nucleic acids using Polymerase Chain Reaction (PCR) as well as other areas of liquid biopsy and, possibly, mass spectrometry analysis of volatile metabolites can be applied for LC screening. Immunoassay is inexpensive and allows for the analysis of a large number of samples, but the analysis of NSCLC-specific markers, such as CEA (carcinoembryonic antigen), NSE (neuron-specific enolase), CYFRA 21-1 (a soluble fragment of cytokeratin 19 associated with epithelial cell tumors), and SCCA (squamous cell carcinoma antigen), does not have sufficient sensitivity and specificity for early LC detection [14]. The analysis of volatile organic compounds (VOCs) using mass spectroscopy could serve as a screening tool for lung cancer due to its rapidity and technical simplicity [15]. However, there is still not a set of VOC biomarkers that are generally accepted and suitable for diagnosing LC [16], not to mention the possible limitations in terms of the specificity of such tests [17]. Liquid biopsy involves the examination of extracellular fluids, primarily blood, for tumor-specific markers such as cell-free DNA and RNAs, circulating tumor cells, and extracellular vesicles (EVs), as well as other circulating molecular and cellular analytes, which offer promising opportunities for cancer diagnosis [18]. Nucleic acids studied by different variants of PCR and next-generation sequencing (NGS) represent the main diagnostic material of liquid biopsy. Moreover, few tests like the Epi proLung^®^ BL Reflex Assay (Epigenomics AG, Berlin, Germany) [19] or Lepu Medical Methylation Kit (Lepu Medical Technology Co., Ltd., Beijing, China) have a CE-IVD and China NMPA approval (but not FDA). There are also few officially approved NGS panels for the detection of mutations and rearrangements in genes involved in LC development and crucial for the selection of LC-targeted therapy, but they are not intended for use for the early diagnosis of LC [20,21]. The limited number of diagnostic kits and the lack of their widespread use make the search for early LC diagnosis markers and new early LC diagnostic tests an urgent task. Such LC screening tests must be robust and inexpensive, such as liquid biopsy. The tests should not lead to overdiagnosis, detection of cancer at the precancerous stage, or cancer in situ but should reliably detect cancer at the early stages of development, when current protocols of LC curation are to be applied and patients can be cured of LC. In one breakthrough retrospective study, the authors were able to predict the incidence of cancer up to four years before its clinical manifestation [22]. However, it is not yet clear whether such data would improve outcomes or whether the negative consequences of such data would outweigh the positive ones.

In this regard, the issue of early markers involves several key considerations. First, markers found exclusively or predominantly at the early stages are strong candidates for early-stage detection. In contrast, markers specific to later stages can be used in conjugation for accurate tumor staging. Furthermore, markers that appear early and persist throughout cancer progression are prime candidates for early diagnostics as their expression may be independent of tumor stage, remaining consistent as the disease advances.

Other markers may be linked to tumor type or heterogeneity yet still serve as good indicators of early-stage cancer. It is obvious that the markers involved in the main pathways of tumor maintenance, especially those frequently encountered across different phenotypes and heterogeneities, are also strong candidates for the timely diagnosis of NSCLC.

In other words, various markers can be used for timely NSCLC diagnosis; however, markers that occur in the early stages may be more promising since they do not require additional preliminary studies involving groups of patients in the early stages, mainly before verification. Additionally, screening potential markers based on their involvement in non-overlapping regulatory pathways, clustering, frequency of occurrence, and features of analytical systems (specificity and multiplication in one probe) is highly valuable for selecting the most promising markers for a diagnostic panel.

For the timely diagnostics of NSCLC, cell-free microRNA (cf-miRNA) presents a promising tool. These nucleic acids are released by tumor cells into the circulation, making them highly accessible biomarkers in blood plasma and a core component of liquid biopsy for NSCLC detection that can easily be isolated and quantified [23,24]. Tumor-specific microRNAs were found at the early and late stages of NSCLC in numerous studies [25]. While early NSCLC diagnostics relied on a single microRNA [26], later research proposed diagnostic panels comprising dozens of different miRNAs [27]. Significant heterogeneity in miRNA selection criteria and experimental methodologies has led to a lack of consensus among studies, with few miRNAs being independently validated.

This review addresses the following question: “Which circulating cell-free miRNAs show the strongest diagnostic potential for early-stage NSCLC?”. To answer this, we provide a novel integrative analysis of validated studies. We analyze data on miRNA occurrence across stages, functional roles in tumorigenesis, normalization methods, and diagnostic performance, based on multiple panels, to critically evaluate and identify the most promising miRNA sets for early detection panels.

## 2. Cell-Free miRNAs as Promising Biomarkers for Early-Stage NSCLC

A distinct spectrum of cell-free miRNAs (cf-miRNA) has been characterized in the circulating blood of patients with early-stage NSCLC. A subset of these cf-miRNAs was initially discovered through large-scale screening methods like microarray platforms or NGS. However, the presence of the majority of candidate cf-miRNAs has been confirmed using more targeted methods such as PCR. The latter were selected from large-scale data as well as data on miRNA expression in different cell types or synthetic data obtained by bioinformatic methods or via looking through a database [28]. In principle, different analytical methods should generate consistent data on microRNA identity and abundance. However, the results of different analytical platforms often converge. A major contributing factor is the absence of universal protocols, which leads to significant variability in the efficacy and reproducibility of miRNA isolation, analysis, and data normalization.

While miRNAs can be isolated from various biological sources like sputum and bronchoalveolar lavage (BAL), this review focuses exclusively on blood-based samples due to their universality and minimally invasive nature, which are critical for widespread early-stage screening [29]. Here, we will discuss the list of miRNAs found in early NSCLC patient’s blood. microRNAs will be ranked by frequency of occurrence in different studies (Table 1), and the key features of the study, including sample type (plasma, serum or microvesicles), the analytical method used, and the normalization method, will be mentioned for subsequent discussion of the applicability of the marker in further sections of the review (Table 2).

The microRNAs miR-145 and miR-21 are among the most frequently identified biomarkers in studies of NSCLC. In particular, miR-21 has been consistently implicated in the carcinogenic process, showing elevated levels especially at early disease stages [30,31,32,33]. Multiple studies employing quantitative RT-PCR with different normalization controls (miR-16, RNU66, miR-197) have demonstrated that plasma and tissue levels of miR-21 tend to increase in NSCLC vs. healthy donors [30,31,32]. However, contradictory results exist regarding whether miR-21 expression correlates precisely with NSCLC stage. For example, it was found that patients with advanced stages (III–IV) exhibit significantly higher plasma miR-21 expression than those in early stages (I–II), and elevated tissue expression correlates with disease development even within stage I patients across diverse cohorts. Conversely, another study showed higher miR-21 serum levels in stage I patients compared to the controls; it found no significant distinction between early and later stages, indicating that miR-21 expression stabilizes after initial increase [32]. Overall, these converging findings suggest that miR-21 is a promising biomarker for early NSCLC detection, particularly effective in identifying squamous cell carcinoma [33].

miR-145 is a well-established regulatory molecule and a validated biomarker for its early detection in NSCLC. Findings using TaqMan miRNA assays with RNU48, miR-191, and cel-miR-39 as normalized controls demonstrated that miR-145 expression levels were significantly downregulated in the plasma of NSCLC patients compared to healthy controls [34,35,36]. However, the correlation between miR-145 expression and disease stage presents inconsistent results. While several studies reported no significant difference in the expression levels across the NSCLC stages [34,36,37], others observed a progressive decrease in miR-145 with advancing disease [38]. Despite this discrepancy regarding stages, the consistent downregulation in patients supports the potential of miR-145 as a valuable tool for the early detection of NSCLC.

Researchers frequently studied miR-126 level as well for its diagnostic potential in early-stage NSCLC. In studies utilizing TaqMan miRNA assays normalized to references such as U6 and cel-miR-39, the results indicate that miR-126 is generally downregulated in NSCLC patients compared to healthy controls, with this downregulation becoming more pronounced in advanced stages [39]. Additionally, miR-126-3p was evaluated in the serum of NSCLC patients in the early stages and demonstrated an independent prognostic role in SCC by significantly associating with disease-free survival (DFS); however, no such significant association was observed in the AD cohort, highlighting a histology-specific function for this miRNA [40].

Several reports have investigated the roles of miR-223, miR-20a, and miR-155 as potential diagnostic biomarkers for early-stage NSCLC. They used quantitative real-time PCR (RT-qPCR) with U6 snRNA for normalization. The results consistently showed that the expression levels of these miRNAs were elevated in patients with early-stage NSCLC compared to healthy controls [33,41,42]. Furthermore, the level of miR-155 was found to increase progressively with advancing disease stage, suggesting its potential utility not only in diagnostics but also in monitoring disease progression [43].

Numerous miRNAs have been reported in two or more independent studies as promising diagnostic biomarkers for detecting early-stage NSCLC. The downregulation of let-7 in NSCLC is a hallmark event linked to tumor initiation, progression, and poor prognosis, making it a compelling candidate for diagnostic and prognostic biomarker development [44]. One study reported the decreased expression levels of let-7a in stage I–II NSCLC patients [45]. In a separate study, Let-7b expression levels varied significantly between NSCLC patients with stage IA/B and the controls by measuring serum and plasma levels with Taq-man miRNA assays using cel-miR-54 and cel-miR-238 for normalization [46]. A study on the miR-200 family in NSCLC demonstrated that miR-200a, miR-200b, miR-200c, and miR-429 were significantly downregulated, while miR-141 was upregulated in NSCLC tissue samples in comparison with the control [47]. Separately, miR-125b was found to be elevated in stage I AD patient samples [48]. In contrast, miR-146a was downregulated in cancer tissues overall but was expressed at significantly higher levels in stage I–II samples compared to stage III–IV samples [49]. A comprehensive summary of these and additional findings is provided in Table 2.

**Table 2 ijms-26-12035-t002:** List of miRNAs validated for detection of early-stage NSCLC.

miRNA	Study Model	Sample Type	Analytical Technique	Normalization	Ref.	Regulation	Biological Processes Regulated	Target	Expression Changes	Ref.
**miR-21**	63 NSCLC patients. Stages: I–IV Mouse model	Plasma	RT-qPCR	miR-16	[31]	↑	Cell cycle DNA repair Apoptosis Angiogenesis Proteolysis Cell adhesion MAPK/ERK pathway TGFβ pathway G-protein pathway cell growth and resistance to apoptosis	STAG2, KIF6 MSH2, FANCC, CHD7 PDCD4, APAF1, STAT3, MALT1, SGK3 SOS2, JAG1, MAP3K1, STAT3 WWP1 CCL1, MATN2, TGFBI, VCL MAP3K1, STAT3, SOS2, NKIRAS1, SPRY1, SPRY2 BMPR2, SMAD7 SOS2, TIAM2, GPR64, KRIT1 PTEN	↑ in stage I compared to control group. ↑ as NSCLC progressed.	[31,32,43,50,51]
	Retrospective analysis of three cohorts included 317 NSCLC patients. Stages: I–III	Tissue samples	TaqMan miRNA assay	RNU66	[30]					
**miR-145**	80 NSCLC patients. Stages: I–IV	Plasma, Tissue samples	TaqMan miRNA assay	RNU48	[34]	↓	Proliferation, EMT, Migration, Invasion Metastasis, Immune evasion Cell cycle Apoptosis Tumor growth	GOLM1, RTKN, SOX9 CXCL3 c-Myc Caspase-3/-9, MTDH EGFR, NUDT1	Study A: ↓ in all patients, regardless of clinical stage [36]. Study B: ↓ as NSCLC progressed [38].	[34,52,53,54,55,56]
	70 paired normal and NSCLC tissues. Stages: I–III	Tissue samples	TaqMan miRNA assay	miR-191	[35]					
**miR-126**	45 NSCLC patients. Stages: I–IV	Serum, Exosomal, Exosomal-free	TaqMan miRNA assay and RT-qPCR	U6 cel miR-39	[39]	↓	Angiogenesis VEGF-A/VEGFR-2/ERK PI3K/AKT signaling mTOR signaling EMT Invasion	EGFL7, IRS-1, Crk, VEGFA PIK3R2 LAT1 SOX11, PLOD2	↓ in advanced stage patients compared to early-stage patients [39].	[57,58,59]
**miR-126-3p**	182 NSCLC patients. Stages: IA–IIIA	Serum	RT-qPCR	cel-miR-39	[40]					
**miR-155**	50 NSCLC patients and 50 healthy volunteers. Early stages	Serum	RT-qPCR	Mean of CT of all healthy controls	[42]	↑	Tumor growth Metastasis Migration Invasion	PTEN, SOCS1, SOCS6 Smad2	↑ as NSCLC progressed.	[43,60]
	80 paired normal and NSCLC tissues. Stages: I–III	Tissue samples	RT-qPCR	U6	[43]					
**miR-205**	265 NSCLC patients. Stage: I	Tissue samples	TaqMan miRNA assay	RNU6B	[61]	↑	Growth metastasis EMT Apoptosis NF-κB signaling Proliferation	PTEN, TP53INP1 Cripto-1 APBB2 EGFR	↑ as NSCLC progressed.	[49,61,62,63,64]
**miR-150**	171 NSCLC patients. Stages: I–II	Serum	Stem-loop array reverse transcription PCR (SLA-RT-PCR)	U6 snRNA	[65]	↑	Proliferation Metastasis	SRCIN1, P-STAT3, ROS, EPG5 FOXO4	↑ as NSCLC progressed.	[65,66,67,68,69]
**miR-141**	155 NSCLC patients. Stages: I–III	Tissue samples	TaqMan miRNA assay	RNU44 RNU48	[47]	↑	Angiogenesis	KLF6, VEGFA	Study A: no correlations between the expression level and TNM stage [70]. Study B: ↑ as NSCLC progressed [71].	[47]
**miR-146a**	101 NSCLC patients. Stages: I–IV	Tissue samples	RT-qPCR	RNU6B	[49]	↓	Proliferation EMT Migration Survivability	EGFR, TNF-α, NF-κB and MEK-1/2, and JNK-1/2. Notch2 TRAF6	↓ in advanced stage patients compared to early-stage patients.	[49,72,73]
						↑				
**miR-223**	75 NSCLC patients and 111 tumor-free controls. Stages: I–II	Serum	droplet digital PCR (ddPCR)	UniSp6/cel-miR-39-3p	[74]	↓	PI3K/AKT pathway Proliferation and invasion Migration	EGFR IGF-1R NLRP3, E2F8	↓ in advanced stage patients compared to early-stage patients.	[74,75,76,77,78]
	31 NSCLC patients. Stages: I–IV 3 NSCLC cell lines. Mouse model.	Serum	Real-Time PCR	U6 snRNA	[78]					
**miR-34b/c**	140 AC patients. Stages: I–II 15 human lung AC cell lines.	Tissue samples	RT-qPCR	RNU48	[79]	↓	Tumor growth Cell cycle Apoptosis EMT	TP53 CDK4, CDK6, CCND1 BCL2, SIRT1 Zeb1	N/A	[80]
**miR-183**	33 AD patients. Stage: I 2 NSCLC cell lines.	Tissue samples	RT-qPCR	U6 snRNA	[81]	↑	mTOR regulation Proliferation, migration and cell cycle Metastasis	SESN1 PTEN, FOXO1 LOXL4	N/A	[81,82,83,84]
**miR-1246**	105 NSCLC patients, 50 patients with NMRD and 50 healthy volunteers. Stages: I–IV	Serum	RT-qPCR	cel-miR-39	[85]	↑	Stemness Metastasis, Wnt/β-Catenin Pathway Radioresistance	TRIM17 CPEB4, GSK-3β DR5	↑ in advanced stage patients compared to early-stage patients.	[85,86,87,88,89]
**miR-328-5P**	86 NSCLC patients and 24 healthy donors. Stages: I–IV.	Peripheral blood cells	TaqMan miRNA assay	RNU38B, RNU58A	[90]	↑	Migration	PRKCA, IL-1beta, c-Raf1, LOXL4	N/A	[90,91]
**miR-328-3P**						↓	Genomic stability	H2AX	↓ as NSCLC progressed.	[92]
**miR-200**	168 NSCLC patients and 128 patients with benign lung nodules. Stages: I–II	Plasma (EV-derived)	RT-qPCR	cel-miR-39-3p	[93]	↓	EMT through Notch signaling	Notch ligand Jagged1 and Jagged2, Flt1 PTEN, ABCA1	Stage II patients exhibited the highest expression level compared to the other stages.	[94,95,96,97,98,99,100,101,102]
**Let-7b**	220 NSCLC patients and 220 healthy controls. Stages: IA–IIB	Plasma	TaqMan miRNA assay	Cel-miR-54 Cel-miR-238	[46]	↓	MAPK/ERK pathway Immune response	BRF2 PD-L1	N/A	[103,104,105]
		Serum								
**miR-4732-5p**	18 AD patients and 18 BPN patients. Stage: I.	Serum (EV-derived)	RT-qPCR	miR-20a	[106]	↓	Migration EMT	TSPAN13 XPR1, PI3K/Akt/GSK3β/Snail pathway	↑ as NSCLC progressed.	[107,108]
**miR-374a**	38 NSCLC patients and 27 Heathy controls Stages: IA–IIB	Tissue samples	TaqMan miRNA assay	RNU48, miR-16 and miR-26b	[109]	↓	Metastasis Proliferation	γ-adducin NCK1	N/A	[110,111]

↑ upregulation/higher expression of miRNA. ↓ downregulation/lower expression of miRNA.

## 3. Blood-Based microRNA Biomarkers: Functional Implications in NSCLC

As was mentioned, miR-145, functioning as a key tumor suppressor, is frequently downregulated in the plasma of early-stage NSCLC patients as it acts as a pivotal regulator of proliferation, apoptosis, and migration of tumor cells. miR-145 directly targets *EGFR* (Epidermal Growth Factor Receptor) and *NUDT1* (Nudix Hydrolase 1), thereby inhibiting downstream oncogenic signaling pathways, mainly *EGFR*/*PI3K*/*AKT*, and suppressing tumor growth [52]. Moreover, miR-145 suppresses the cell cycle through inhibiting *c-Myc* and *CLR4*, which leads to the deactivation of the *c-Myc*/*eIF4E* pathway, downregulation of cell cycle regulators such as *Cyclin D1*, and arresting the cell cycle [53]. Additionally, the metastatic potential of NSCLC is suppressed by miR-145 through its inhibition of the *Oct4*-mediated *Wnt*/*β-catenin* signaling pathway and EMT [112]. miR-145 also promotes the apoptosis process in NSCLC cells by activating the caspase cascade primarily through targeting *caspase-3* and by modulating the Bcl-2 family [113]. This regulation of multiple critical pathways highlights the significance of miR-145 as a potential timely biomarker in early-stage NSCLC.

miR-21 plays a central oncogenic role in the development of NSCLC, particularly from the early stages of the disease. Its regulation of key targets such as *PTEN* and *PDCD4* leads to activation of the *PI3K*/*AKT*/*mTOR* signaling pathway, enhancing tumor cell proliferation and disrupting normal differentiation [50,114]. Additionally, it was suggested that miR-21 influences the tumor microenvironment by modulating antioxidant enzymes like *SOD2* and *GPx1*, as well as signaling suppressors *SOCS1* and *SOCS6*, thereby promoting conditions favorable for tumor growth [43,115]. A critical aspect of NSCLC pathogenesis is the reciprocal regulation between miR-21 and the hypoxia-inducible factor *HIF-1α*, which drives metabolic adaptation by increasing glycolysis through *HIF-1α*-induced enzymes and simultaneously supporting angiogenesis via the upregulation of vascular factors such as *VEGF*, as shown in vitro [116,117,118].

This coordinated regulation of metabolism and blood vessel formation facilitates tumor progression and contributes to therapy resistance. Elevated *HIF-1α* levels correlate with advanced tumor stage and poor clinical outcomes, emphasizing the clinical importance of miR-21’s interaction with hypoxia pathways in NSCLC growth. Overall, these findings highlight miR-21 as a multifaceted regulator that orchestrates the key signaling and metabolic pathways driving NSCLC progression, making it a promising target for early intervention and therapeutic strategies.

miR-126 is a critical tumor suppressor and master regulator of angiogenesis in lung cancer. Several studies proposed that miR-126 functions by directly inhibiting key pro-angiogenic targets, including *EGFL7*, *IRS-1*, *Crk*, *VEGF-A*, and, most importantly, the *LAT1*-mediated *mTOR* signaling pathway [57,58]. Additionally, miR-126 is responsible for inducing the programmed cell death pathway and inhibiting the metastasis of tumor cells by targeting the *VEGF-A*/*VEGFR-2*/*ERK* and *PI3K*/*AKT* pathways. These cascades promote essential steps of angiogenesis, including endothelial cell proliferation and tube formation, which results in increased tumor microvessel density. Overall, miR-126 represents a central node in the angiogenic switch and a potential therapeutic and diagnostic target in early-stage NSCLC [57,119].

In early-stage NSCLC, miR-155 plays opposing roles depending on lung cancer cell type. In AD, one study reported that it acts as an oncogene by suppressing *SOCS1* and *PTEN* tumor suppressors, leading to activated *JAK*/*STAT* and *PI3K* signaling pathways that drive tumor growth and correlate with poor patient outcomes [43]. Conversely, in SCC, miR-155 can function as a tumor suppressor by inhibiting *SMAD2* and reducing cancer metastasis, associated with better prognosis in certain patients. This tissue-specific duality complicates its use as a biomarker but highlights its importance in NSCLC biology [60].

miR-205 acts as an oncogene in NSCLC by promoting tumor growth, metastasis, and chemoresistance of tumor cells through targeting *PTEN* [62]. Additionally, miR-205-3p promotes progression and inhibits apoptosis of the NSCLC cells by targeting *APBB2* (Amyloid β Precursor Protein-Binding Family B Member 2), which was associated with worse prognosis in AD patients [63]. Conversely, it was shown that low expression of miR-205 was a negative prognostic indicator in early-stage disease. It identified an inverse correlation between miR-205 and the oncoprotein Cripto-1 (*CR-1*), which, in turn, activates the pro-tumorigenic pathways (*SRC* and *SMAD*) and promotes the EMT pathways, thereby driving metastasis and resistance to targeted therapies such as *EGFR*-TKIs [61].

The antitumor activity of miR-146a is characterized by suppressing proliferation and promoting apoptosis of NSCLC cells by inhibiting *EGFR* and the downstream pathways (*ERK-1/-2*, *AKT*, and *STAT*). By targeting the *IRAK-1* and *NF-kB* signaling pathways, miR-146a reduces tumor cell motility and metastasis [49].

The let-7 family of microRNAs, mainly let-7a, -7b, -7c, and -7g, is one of the most studied tumor suppressor miRNA families in lung cancer, particularly in the context of cancer initiation. Its loss removes a critical brake on cell proliferation (*MAPK*/*PI3K*), stemness (*Wnt*/*β-catenin*), and pro-tumorigenic inflammation (*STAT3*/*NF-κB*) [120]. Therefore, the expression level of let-7 miRNAs serves as a powerful biomarker: high let-7 activity suppresses these pathways and correlates with a favorable prognosis, while low let-7 activity allows for pathway hyperactivation and predicts aggressive disease and poor survival [121]. The profound regulatory function of the let-7 family makes its members significant targets for future NSCLC early detection.

The miR-34b/c cluster is located on chromosome 11q23 and regulated by p53. It exerts significant tumor suppressor activity in early-stage NSCLC [80]. miR-34 family members induce multiple tumor-suppressive effects, including the inhibition of cell migration, cell cycle arrest, and promotion of apoptosis by directly targeting key oncogenes such as *CCNE1*, *EGFR*, and *SIRT6* [122]. Furthermore, it inhibits tumor growth and metastasis through the upregulation of *PTEN* and repression of *YY1*. In parallel, miR-34a, a related family member, significantly reduces cellular proliferation, induces cell cycle arrest, and promotes senescence in lung cancer cells via direct modulation of the E2F transcription factors E2F1 and E2F3 [123].

Analysis of miRNAs targets in early-stage NSCLC has identified key signaling pathways, mainly *PI3K*/*Akt*/*mTOR*, *NF-kB*, *EGFR*, *JAK*-*STAT*, and Notch, as being frequently implicated in carcinogenesis. These pathways primarily drive tumor development by regulating critical processes such as cell cycle progression, proliferation, EMT, migration, invasion, angiogenesis, stemness, and apoptosis, as described in Figure 1. Considering their regulatory functions, the aforementioned miRNAs could be used as additional biomarkers to guide targeted therapy. For instance, the *mTOR* pathway is directly inhibited by Rapamycin and its analogs [124]. Other specific low-molecular-weight kinase inhibitors include Erlotinib targeting *EGFR* [125] and Ruxolitinib targeting *JAK1/2* [126]. Furthermore, high-molecular-weight monoclonal antibodies are generally employed, such as Bevacizumab targeting the *VEGF* ligand to inhibit angiogenesis [127] and Atezolizumab targeting the PD-L1 ligand on tumor cells to enhance immune response [128]. A synergetic approach combining such targeted agents should be further investigated to potentially limit cancer progression.

## 4. Multi-miRNA Panels: Improving Early NSCLC Detection

The data described demonstrates that microRNAs are emerging as powerful tools in the management of early-stage NSCLC. Considering diagnostic efficacy as well as the variability in miRNA expression depending on tumor type and individual patient features, specific miRNA panels (rather than single miRNAs) are required for reliable cancer diagnostics [25]. The selection of miRNAs for these panels is a precise process rooted in high-throughput sequencing and microarray studies that compare miRNA expression profiles between NSCLC tumor tissue and matched healthy tissue. The criteria for selecting miRNA panels emphasize differential expression, stability, functional relevance, and robust performance metrics. The development process involves discovery, validation, panel construction, clinical evaluation, and blinded testing, supported by bioinformatic analysis. These steps ensure that miRNA panels are reliable, reproducible, and clinically useful for non-invasive diagnostics [129]. miRNAs are selected based on their significant dysregulation (up- or downregulation) in disease states compared to healthy controls. Moreover, they must be stable and reliably detectable in liquid biopsies (e.g., serum, plasma). Furthermore, the chosen miRNAs should be involved in several key cancer pathways (e.g., proliferation, apoptosis, metastasis). In addition, it is desirable that there be no correlation between the expression of microRNA biomarkers and the patient’s gender and age, as well as physiological state (hormonal status, dietary pattern, etc.) [130]. Therefore, the miRNA panels should be evaluated based on multiple factors such as sensitivity, specificity, area under the curve (AUC), positive likelihood ratio (PLR), negative likelihood ratio (NLR), and diagnostic odds ratio (DOR) [131].

Despite their promising results, the panels reviewed here share common limitations that affect their real-world reliability. Some studies, like the one performed by Zhong et al. (2021) [132], used relevant control groups (including benign nodules) but were limited by small or non-diverse cohorts. Others, like the one performed by Ying et al. (2020) [133], enrolled large, multi-ethnic populations but did not test specifically against benign conditions, since distinguishing cancer from benign nodules is the main challenge in screening. Blinded validation was also inconsistently applied. To build panels that are both reproducible and clinically useful, future work must prioritize diverse patient groups, include benign disease controls, and use blinded evaluation. Without these steps, the findings will remain difficult to compare or translate into practice.

A key challenge for miRNA panels in NSCLC detection is ensuring high specificity as many individual miRNAs (e.g., miR-21) are dysregulated across various cancer types, potentially leading to false positives. While miRNA panels show strong sensitivity for lung cancer, they must be carefully validated to differ between NSCLC and other malignancies, resulting in an accurate disease-specific detection [134]. However, this does not imply that different cancer types share an identical global cf-miRNA expression profile. A promising solution to this problem is paired normalization within a miRNA panel. This approach involves analyzing the ratios between specific miRNA pairs rather than relying solely on the absolute levels of individual miRNAs. This can significantly improve the accuracy of disease-specific detection and differentiate NSCLC from other malignancies [135].

The growing body of evidence demonstrates that miRNA panels hold great promise as non-invasive diagnostic tools for early-stage NSCLC. The rigorous selection process, which integrates high-throughput sequencing, functional pathway relevance, and advanced bioinformatic analyses, ensures that candidate miRNAs exhibit both biological significance and clinical utility. Multiple studies, some mentioned in Table 3, have validated distinct miRNA signatures with high diagnostic accuracy, achieving impressive AUC values often exceeding 0.85 to 0.97, along with strong sensitivity and specificity across diverse patient cohorts and NSCLC subtypes. The composite score for each diagnostic panel was calculated using the following formula: AUC + (specificity/100) + (sensitivity/100). This metric provides an integrated assessment of the panel’s overall performance. For instance, the panel proposed by Zhong et al. (2021) achieved a score of 2.385 [132], which is lower than the score of 2.867 [135] attained by the panel from Wang, Y. et al.’s study (2016) [136].

Notably, subtype-specific panels targeting AD and SCC have improved diagnostic granularity by capturing unique molecular profiles within heterogeneous NSCLC populations.

The use of liquid biopsies such as serum and plasma samples for miRNA detection emphasizes the clinical feasibility of these panels for routine screening and early diagnosis. However, variability in performance metrics across studies highlights the urgent need for standardization of sample processing, normalization controls, and diagnostic thresholds. While miRNA panels have shown superior diagnostic performance compared to single miRNA markers, challenges remain in establishing universally accepted panels suitable for clinical application. Large-scale, prospective, multicenter studies are essential to refine panel composition and validate them. The validation of pre-analytical steps (blood processing, isolation of microvesicles, isolation and processing of miRNAs) and standardization of the analytical system, including universal spike-in control and internal control of few stably expressed cf-miRNAs, are demanded for the successful development of NCLC miRNA-based diagnostics.

**Table 3 ijms-26-12035-t003:** List of blood-based miRNA panels validated for early-stage NSCLC diagnostics.

Panels	miRNAs	Study Cohort	Sample Type	AUC	Sensitivity	Specificity	Panel Score	Ref.
**Zhong, Y. et al. (2021)**	miR-520c-3p, miR-1274b	207 NSCLC patients, 168 healthy controls, 31 benign nodule patients	Serum and plasma	0.823	82.3	73.9	2.385	[132]
**Wang, P. et al. (2015)**	miR-125a-5p, miR-25, miR-126	94 NSCLC patients and 48 stage III–IV NSCLC patients and 111 healthy controls	Serum	0.936	87.5	87.5	2.686	[137]
**Zheng, D. et al. (2011)**	miR-155, miR-197, miR-182	74 NSCLC patients and 68 healthy controls	Plasma	0.9012	81.33	86.76	2.5821	[138]
**Lv, S. et al. (2017)**	miR-146a, miR-222, miR-223	180 AD patients and 180 healthy controls	Serum	0.951	84.35	90.83	2.7028	[139]
**Peng, H. et al. (2016)**	miR-1254, miR-485-5p, miR-574-5p	Training set: 36 NSCLCs vs. 36 controls validation set: 120 NSCLCs and 71 controls	Serum	0.844	93.3	73.2	2.509	[140]
**Aiso, T. et al. (2018)**	miR-145-5p, miR-20a-5p, miR-21-5p	56 NSCLC patients and 26 healthy controls	Serum	0.893	85.7	80	2.55	[36]
**Ying, L. et al. (2020)**	let-7a-5p, miR-1-3p, miR-1291, miR-214-3p, miR-375	744 NSCLC patients and 944 healthy controls	Serum	0.935	82.9	90.7	2.671	[133]
**Yang, X. et al. (2019)**	miR-146b, miR-205, miR-29c and miR-30b	128 NSCLC patients and 30 healthy controls	Serum	0.96	95.31	82.98	2.7429	[141]
**Poh, K.C. et al. (2025)**	miR-196a-5p, miR-1268, miR-130b-5p, miR-1290, miR-106b-5p, miR-1246	82 NSCLC patients and 123 healthy controls	Serum	0.989	92.1	97.5	2.885	[13]
**Wang, Y. et al. (2016)**	miR-532, miR-628-3p, and miR-425-3p	201 early-stage and 25 late-stage AD patients and 43 patients with lung benign disease and 178 healthy controls	Plasma	0.976	90.2	98.9	2.867	[136]
**Leng, Q. et al. (2017)**	miR-21, 210, and 486-5p	92 LC patients and 88 cancer-free smokers	Plasma	0.85	75.5	85.3	2.458	[142]
	miR-126, miR-145, miR-210, and miR-205-5p			0.96	91.5	96.2	2.837	
**Abdipourbozorgbaghi, M. et al. (2024)**	miR-9-3p, miR-96-5p, miR-147b-3p, miR-196a-5p, miR-708-3p, miR-708-5p, miR-4652-5p	78 NSCLC patients and 44 healthy controls	Plasma	0.85	83	78	2.46	[143]
	miR-130b-3p, miR-269-3p, miR-301a-5p, miR-301b-5p, miR-744-3p, miR-760, miR-767-5p, miR-4652-5p, miR-6499-3p			0.88	92	73	2.53	

## 5. Conclusions and Perspectives

Comparing measurements of circulating miRNAs from different studies can be challenging as they can be affected by pre-analytical and biological factors. For instance, serum preparation involves coagulation, which induces the lysis of red blood cells, resulting in the release of intracellular RNA, thereby altering the profile of circulating extracellular miRNAs. Consequently, plasma is often favored over serum for miRNA studies as it avoids coagulation and reflects normal circulating miRNAs. Moreover, biological differences such as gender-related variations in circulating miRNA levels should be considered [144]. Besides blood-based samples, respiratory fluids such as sputum and BAL fluid offer a significant advantage for lung cancer-specific miRNA profiling. These samples provide a more localized snapshot of the tumor microenvironment, potentially enriching for lung-derived miRNAs and reducing background noise from systemic sources, but they are less universal and more difficult to standardize [145].

The search for diagnostic microRNAs for detecting lung cancer in the early stages and the assembly of microRNA panels is complicated by the limited number of patients with reliably diagnosed early-stage lung cancer. According to the data in Table 2, the majority of early-stage cancer biomarkers are also present in its late stages. This means that the search for markers can be carried out using plasma samples from patients with late-stage cancer, followed by verification of the most promising diagnostic microRNAs on cf-miRNA samples from patients at stages I and II.

Simultaneously, when selecting miRNA biomarkers for diagnostic or prognostic purposes, it is necessary to verify that their expression is specific to the physiological state and not confounded by other patient variables. Evidence shows that global miRNA expression in blood can be influenced by age [146,147] and hormonal status [148]. Moreover, the presence of the disease itself can shift the physiological miRNA profile [147]. Therefore, it is essential to ensure that the expression of the selected miRNA biomarkers does not correlate with patient characteristics.

The normalization step is critical when quantifying circulating miRNA expression levels as it is essential for differentiating the true biological signals from non-biological variations. Typical normalization methods use endogenous small non-coding RNAs such as miR-16, U6, or SNORD RNAs, which can be poorly suited for fluid-based samples and not suitable for normalization circulating miRNA data [144,149]. The introduction of exogenous spike-in controls for normalization is also used. This involves adding synthetic miRNAs (e.g., from *C. elegans*) to each sample at the start of RNA extraction and helps control the technical variations in RNA extraction and amplification efficiency [142,150]. For further refinement, data-driven approaches like global mean normalization (using the average of all detected miRNAs) or the mean-endo method (using the geometric mean of a pre-validated panel of stable endogenous miRNAs) can be applied to minimize bias and improve accuracy [151]. However, better normalization methods have been developed, such as pairwise normalization. This advanced strategy abandons the concept of a single reference gene. Instead, it uses the ratio between two miRNAs as a stable, unit-less measurement which cancels out the majority of the technical and non-biological variability that affects both miRNAs in a pair [152].

So far, adapting a universal reference miRNA for blood-based studies is not feasible due to significant biological and technical variability. This variability arises from differences in how miRNAs are packaged and released into circulation, changes in miRNA expression related to disease states, and inconsistencies in sample preparation methods. Therefore, the focus should shift from searching for a single universal normalizer to adopting a standardized method where each study validates its own set of stable reference miRNAs specific to its experimental conditions and cohort. This approach prioritizes reliable and comparable results within a study over broad generalization [153].

The performance variations among panels can be attributed to several factors, including the biological samples used (serum, plasma, or sputum), the selection method for miRNA candidates, the population characteristics (ethnicity, smoking status, cancer stage, physiological state), and the analytical techniques employed for miRNA quantification. For instance, the integrated approach using both sputum and plasma miRNAs demonstrated great performance (87% sensitivity and 89% specificity) compared to panels using single biofluid sources. This enhancement highlights the complementary nature of different biomarker sources and the potential advantage of multi-source sampling strategies [154]. Furthermore, the number of miRNAs in each panel appears to influence diagnostic performance, though not always linearly. While some studies found that smaller panels (3–5 miRNAs) achieved excellent performance [139], others developed more comprehensive signatures (6–9 miRNAs) to maximize diagnostic accuracy [13]. Therefore, the variation suggests that beyond mere quantity, the biological relevance and functional coordination of selected miRNAs within pathways crucial to lung carcinogenesis are paramount considerations for developing effective diagnostic panels [155].

The integration of miRNA biomarkers with current low-dose computed tomography (LD-CT) screening protocols represents the most promising application for improving early NSCLC detection. Research demonstrates that combining miRNA signatures with radiologic imaging significantly enhances diagnostic performance. For instance, a study incorporating six serum miRNA biomarkers with nodule size achieved remarkable AUC values between 0.96 and 0.99, with sensitivities of 92–98% and specificities of 93–98% [13]. This represents a substantial improvement over either approach alone, highlighting the synergistic effect of combining molecular and imaging biomarkers.

While miRNAs such as miR-20a, miR-21, miR-145, and miR-223 are frequently validated individually for the early detection of NSCLC, multi-miRNA panels often demonstrate better diagnostic value. Our analysis of 14 independent panels, listed in Table 3, aimed to identify the most robust biomarkers. To achieve this, for each miRNA, we calculated its mean composite score by averaging the panel scores of all panels in which this miRNA appeared. This provided a metric to rank the miRNAs by their average diagnostic performance across studies. This approach identified miR-1246, miR-1290, miR-130b-5p, miR-532, miR-205, miR-29c, miR-126b, and miR-222 as top-performing candidates (Figure 2) from the chosen panels. These miRNAs can be recommended for inclusion in future panels for the timely detection of NSCLC as they contributed to ensuring high accuracy across the chosen studies.

Despite the promising diagnostic performance of multi-miRNA panels in research studies, their translation into clinical diagnostics faces significant challenges, with no panel currently holding FDA or CE approval. This can be due to lack of standardization in pre-analytical and analytical methods, including sample collection, processing, and data normalization, which impairs reproducibility and comparability across studies [156,157]. As mentioned earlier in the article, many candidate panels lack robust validation in large-scale, prospective, and ethnically diverse multicenter cohorts, which is essential to prove generalizability and clinical utility. Finally, the technical complexity and high cost of implementing advanced discovery platforms in routine clinical laboratories present a significant practical challenge [158]. To overcome these challenges, we recommend shifting the focus from individual miRNA biomarkers to the validation of multi-miRNA signatures within prospective, multicenter studies. This requires prospective, blinded validation studies to be performed across multiple centers and diverse populations using standardized methodologies. Such an approach represents a critical first step toward translating these panels into clinical diagnostics.

## Figures and Tables

**Figure 1 ijms-26-12035-f001:**
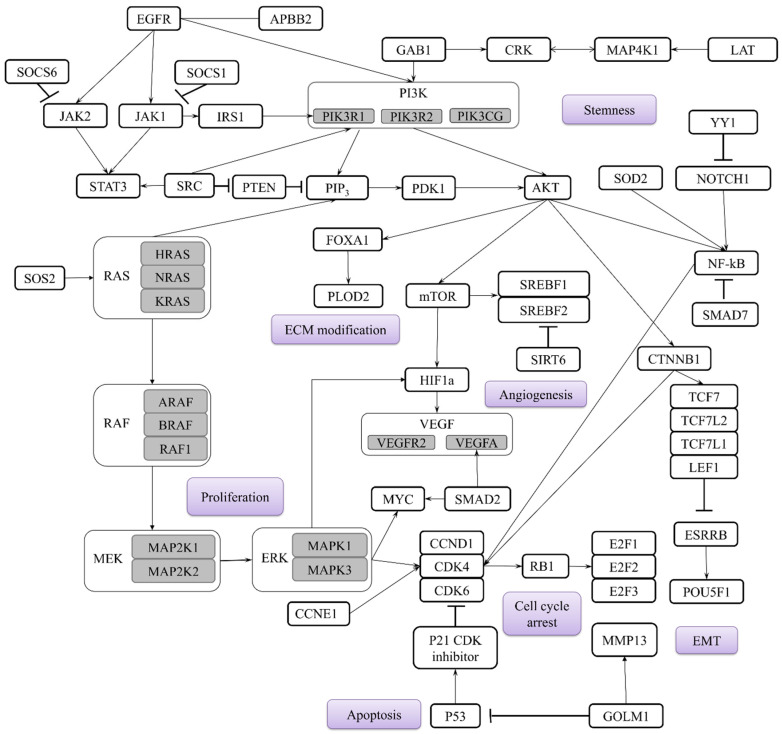
Main genes targeted by miRNAs in early-stage NSCLC and main biological processes involved in the carcinogenesis highlighted in purple. Subtypes of the main genes are highlighted in grey.

**Figure 2 ijms-26-12035-f002:**
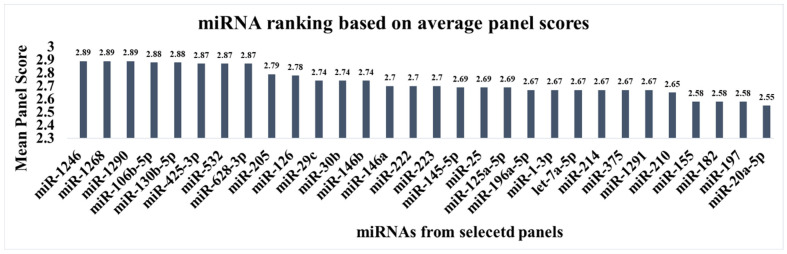
Ranking of miRNAs by their average performance score in the selected diagnostic panels.

**Table 1 ijms-26-12035-t001:** Most frequently reported miRNAs in early-stage NSCLC detection studies.

Number of Validating Studies	miRNAs
≥11	miR-145
≥10	miR-21
≥5	miR-126
≥4	miR-20a, miR-223, miR-155, miR-205, miR-210, miR-1246
≥3	miR-150, miR-205, miR-574-5p, miR-146a, miR-486
≥2	let-7a, Let-7b, miR-29c, miR-34b, miR-125a-5p, miR-125b, miR-141, miR-182, miR-183, miR-222, miR-429, miR-886, miR-1254, miR-1290, miR-106b
≥1	More than 66 different miRNAs have been reported across the studies

## Data Availability

No new data were created or analyzed in this study. Data sharing is not applicable to this article.

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
