# Peer review of "Blood-Based miRNA Panels for Timely Detection of Non-Small-Cell Lung Cancer: From Biomarker Discovery to Clinical Translation"

_ijms, 2025, doi:10.3390/ijms262412035_

Round 1
Reviewer 1 Report
Comments and Suggestions for Authors
Dear Authors,
Thank you for submitting your manuscript entitled “Blood-based miRNA Panels for Timely Detection of Non-Small Cell Lung Cancer: From Biomarker Discovery to Clinical Translation.” The topic is timely and highly relevant, and the manuscript is overall well structured and comprehensive. Below I provide detailed feedback aimed at strengthening scientific rigor, clarity, and clinical relevance.
Your review provides an extensive and well-organized synthesis of current blood-based miRNA biomarkers for early-stage NSCLC. The tables summarizing validated miRNAs (Table 2) and diagnostic panels (Table 3) are highly valuable for readers, and Figures 1 and 2 enhance comprehension of molecular pathways and biomarker ranking.
However, some sections would benefit from clearer definitions, improved methodological consistency, more critical appraisal, and a tighter connection between the identified gaps and future directions. Certain claims require more cautious language or additional justification. In addition, some structural and editorial issues should be addressed for improved readability.
- Line 28–41: The introduction mixes epidemiology, environmental exposures, and biology. Consider reorganizing to clearly define the gap this review intends to fill (e.g., the absence of clinically validated cf-miRNA panels for early NSCLC).
- The review question should be explicitly stated, e.g., “Which circulating miRNAs have been validated for early-stage NSCLC detection, and which combinations show the strongest diagnostic value?”
- Throughout the manuscript you mention differences in normalization, sample type, or analytical platforms.
→ It would strengthen the review to systematically evaluate how these methodological differences affect reproducibility.
→ For example: in Table 2, some studies use U6, others cel-miR-39, others mean Ct—yet their limitations are not explicitly discussed.
- Lines 221–276: The functional summaries of miR-21, miR-126, miR-155, etc., are informative, but often written as definitive mechanistic statements.
Wherever claims are based on single studies or limited cell models, consider adding qualifiers such as “has been suggested,” “was shown in vitro,” etc.
- Lines 299–355: Excellent presentation of panel performance.
However, it would benefit from: - Discussion of population limitations (e.g., Asian vs. European cohorts).
- Whether panels were validated in blinded settings.
- Handling of benign mimickers (e.g., inflammatory nodules, infections), which are common in real-world screening.
- Lines 356–375 describe limitations but do not fully translate them into concrete research priorities.
Clarity and Style
- Line 14: “timely detection strategies” → consider replacing with “strategies enabling detection at curable stages (I–II)”.
- Line 49: “we will concentrate on timely NSCLC diagnostics” → replace “timely” with “early-stage”.
- Line 96–110: Several sentences are long and difficult to follow; consider splitting for clarity.
- Ensure consistent use of:
- NSCLC vs. LC (sometimes mixed without explanation).
- Exosomes / extracellular vesicles (define EVs on first use).
- Table 1: Consider adding the direction of regulation (↑/↓) to improve usability.
- Table 2: Some entries repeat the same target genes across miRNAs—ensure that this is accurate and not a formatting carryover.
- Table 3: The “Panel Score” column is useful but should be explained clearly in the caption.
- Figure 1: Pathway diagram is dense; adding color coding for tumor-suppressor vs. oncogenic miRNAs would enhance readability.
- Figure 2: Please specify in the text how the composite performance score was calculated.
Your manuscript addresses an important field and has strong potential. I recommend major revision, particularly to improve methodological critique, clarify mechanistic interpretations, and strengthen the clinical perspective. Addressing the comments above will significantly enhance the scientific and translational value of the review.
I appreciate your contribution to advancing miRNA-based diagnostics and hope these comments support the refinement of your manuscript.
Author Response
Comment 1: The introduction mixes epidemiology, environmental exposures, and biology. Consider reorganizing to clearly define the gap this review intends to fill (e.g., the absence of clinically validated cf-miRNA panels for early NSCLC).
Response 1: Thank you for this essential suggestion to sharpen the focus of our introduction. We agree with this comment; therefore, we have completely reorganized this section to build a clear, logical narrative the which will help to present the idea of miRNA later on in the upcoming sections
Comment 2: The review question should be explicitly stated, e.g., “Which circulating miRNAs have been validated for early-stage NSCLC detection, and which combinations show the strongest diagnostic value?”
Response 2: We agree. We have now explicitly stated the review question in the introduction as recommended: “Which circulating cell-free miRNAs show the strongest diagnostic potential for early-stage NSCLC?” This question clearly frames the objective of our subsequent analysis.
Comment 3: Throughout the manuscript you mention differences in normalization, sample type, or analytical platforms.
→ It would strengthen the review to systematically evaluate how these methodological differences affect reproducibility.
→ For example: in Table 2, some studies use U6, others cel-miR-39, others mean Ct—yet their limitations are not explicitly discussed.
Response 3: Thank you for your feedback. The primary goal of Sections 2 and 3 was to provide a comprehensive summary of studies validating miRNAs as biomarkers for early-stage lung cancer detection. Considerations regarding normalization and methodological variability were addressed in the limitations section of the conclusion. We tried in this article to maintain the overall focus and scope, ensuring proportional emphasis on the core findings of the reviewed studies. Regarding the normalization idea, we did prepare a separate article which was sent to review in IJMS with the title: “Comparison of Normalization Methods for Selecting a Non-Small Cell Lung Cancer Marker Panel from Circulating miRNA RT-qPCR Data” and now we are answering the reviewers’ comments as well.
Comment 4: The functional summaries of miR-21, miR-126, miR-155, etc., are informative, but often written as definitive mechanistic statements.
Wherever claims are based on single studies or limited cell models, consider adding qualifiers such as “has been suggested,” “was shown in vitro,” etc.
Response 4: We agree with this suggestion and thank you for the careful reading. To more accurately reflect the strength of the evidence, we have revised the text by adding appropriate qualifiers (e.g., "suggested," "proposed," "shown in vitro") to mechanistic statements, particularly those based on single or limited studies.
Comment 5: Excellent presentation of panel performance. However, it would benefit from:
- Discussion of population limitations (e.g., Asian vs. European cohorts).
- Whether panels were validated in blinded settings.
- Handling of benign mimickers (e.g., inflammatory nodules, infections), which are common in real-world screening.
Response 5: We thank you for this valuable suggestion to enhance the critical analysis of the panels. We have expanded the discussion in the relevant section to address these points directly. Specifically, we now:
- Discuss cohort limitations, contrasting studies with large, multi-ethnic populations (e.g., Ying et al., 2020) against those with smaller, less diverse cohorts (e.g., Zhong et al., 2021).
- Note the inconsistent reporting of blinded validation across studies.
- Emphasize that the inclusion of patients with benign pulmonary nodules is a critical yet often missing control for establishing real-world specificity.
This addition strengthens our critique by highlighting that for panels to be reproducible and clinically actionable, future work must optimize cohort diversity, blinded design, and validation against benign mimickers.
Comment 6: Lines 356–375 describe limitations but do not fully translate them into concrete research priorities.
Response 6: We appreciate your comment. However, our intention in the conclusion was to move from stating limitations to proposing possible solutions. The specific recommendations we provided such as adopting pairwise normalization to overcome technical variability and prioritizing studies on integration miRNA with Low-dose computed tomography (LD-CT) were intended as direct research priorities to address the stated methodological and translational challenges.
Comment 7: Line 14: “timely detection strategies” → consider replacing with “strategies enabling detection at curable stages (I–II)”.
Response 7: We appreciate your comment regarding terminology. We chose "timely diagnosis" over "early diagnosis" to address specific practical and conceptual limitations. "Early diagnosis" can imply detection at asymptomatic or in situ stages, which presents major research feasibility issues and potential overdiagnosis concerns.
As for diagnosis at curable stages, new drugs and strategies are, in principle, pushing the boundaries of "curability," while complete remission is difficult to achieve in any case. Therefore, we used the term "timely diagnosis," i.e., diagnosis whose results can be effectively used for the benefit of the patient.
In light of the above, and considering the fact that other reviewers did not comment on this term, we would prefer to retain the term "timely diagnosis" in the text of the article.
Comment 8: Line 49: “we will concentrate on timely NSCLC diagnostics” → replace “timely” with “early-stage”.
Response 8: Thank you and we changed the word “timely” in this line with “early-stage” as recommended.
Comment 9: Several sentences are long and difficult to follow; consider splitting for clarity.
Response 9: Thank you for highlighting the lack of clarity in the chosen lines, we have carefully revised this paragraph to improve readability by splitting the long sentences.
Comment 10: Ensure consistent use of:
- NSCLC vs. LC (sometimes mixed without explanation).
- Exosomes / extracellular vesicles (define EVs on first use).
Response 10: We appreciate your attention to the terminology. The manuscript has been revised to ensure consistent usage: the term "extracellular vesicles (EVs)" is now explicitly defined at its initial introduction, and "LC" has been replaced with "NSCLC" several times when it is needed not to mix between the two terms.
Comment 11: Table 1: Consider adding the direction of regulation (↑/↓) to improve usability.
Response 11: We appreciate the reviewer's suggestion to enhance Table 1. After careful consideration, we have decided to keep the current format. The direction of regulation for individual miRNAs is detailed in Table 2, which is dedicated to functional analysis. Adding this data to Table 1, where some miRNAs (e.g., miR-146a) are reported with opposing directions in the literature, could reduce clarity and create confusion for readers assessing the diagnostic panels.
Comment 12: Table 2: Some entries repeat the same target genes across miRNAs—ensure that this is accurate and not a formatting carryover.
Response 12: Thank you for your careful review. The repeated target genes (e.g., PTEN) across different miRNAs in Table 2 are accurate and reflect the established biology where key tumor suppressor genes are commonly regulated by multiple miRNAs. We have verified each entry against the cited primary literature.
Comment 13: Table 3: The “Panel Score” column is useful but should be explained clearly in the caption.
Response 13: We agree. Thank you for your comment. We added the concept of panel score and how to calculate it in the end of fourth section as following: “The composite score for each diagnostic panel was calculated using the formula: AUC + (Specificity/100) + (Sensitivity/100). This metric provides an integrated assessment of the panel's overall performance. For instance, the panel proposed by Zhong et al. (2021) achieved a score of 2.385 [134] which is lower than the score of 2.867 [135] attained by the panel from Wang, Y., et al. (2016) [135]”.
Comment 14: Figure 1: Pathway diagram is dense; adding color coding for tumor-suppressor vs. oncogenic miRNAs would enhance readability.
Response 14: Thank you for this suggestion. We considered color-coding but concluded it might lead to misinterpretation. As several genes are targeted by miRNAs with opposing roles, and because the network includes both direct and indirect regulation, a single functional color per miRNA could inaccurately represent the pathway's complexity. We believe the current format presents the interactions most clearly.
Comment 15: Please specify in the text how the composite performance score was calculated.
Response 15: We appreciate the suggestion for clarity. The method for calculating the composite performance score in Figure 2 has been added to the manuscript text. It is defined as the mean of the individual panel scores (AUC + Sensitivity/100 + Specificity/100) across all panels that included the respective miRNA.

Reviewer 2 Report
Comments and Suggestions for Authors
1.This research focused on Blood-based miRNA Panels for Timely Detection of Non-Small Cell Lung Cancer: From Biomarker Discovery to Clinical Translation, after check in pubmed, not so many references about this topic(PMID: 29805626 ), this was mean this manuscript was with some Innovation.
2.Whole manuscript contained so much data and contents, but I think some places can be more perfect.
- Early diagnosis is very important for the prognosis of patients with lung cancer. This review summarizes the clinical significance of blood miRNA for early diagnosis of lung cancer.
- Please clarify the focus and importance of this article compared with the published review “Clinical significance of blood-based miRNAs as biomarkers of non-small cell lung cancer. Oncol Lett. 2018 Jun;15(6):8915-8925. doi: 10.3892/ol.2018.8469.”
- Abbreviation should arrange in alphabetical order.
- How to get the Figure 2 results?
7.So many articles about miRNA and cancer were withdraw, so articles related with miRNA was untrustworthy , how do you evaluate the authenticity of your selected literature?
Author Response
Comment 1:
Please clarify the focus and importance of this article compared with the published review “Clinical significance of blood-based miRNAs as biomarkers of non-small cell lung cancer. Oncol Lett. 2018 Jun;15(6):8915-8925. doi: 10.3892/ol.2018.8469.”
Response 1:
Thank you for raising this important point. Our review provides a distinct and updated contribution to the field, complementing and extending the valuable work of Li et al. (2018) doi: 10.3892/ol.2018.8469 in Oncology Letters. The key differentiating and novel aspects of our manuscript are:
- Broadened biomarker scope: While the cited review focused specifically on exosomal miRNAs, our work provides a comprehensive review on cell-free miRNAs (cf-miRNAs) in circulation.
- Updated and comprehensive synthesis: Our analysis incorporates studies published over the subsequent five years, capturing significant advances in the field. This allows us to evaluate a substantially larger and more current body of evidence, reflecting the latest validation studies and methodological developments.
- Panel evaluation and miRNA ranking: We examined specific diagnostic panels, explained why they were chosen, and used their results to rank which miRNAs work best. This helps select the most reliable miRNAs for future tests.
- Normalization strategy analysis: We reviewed the different ways studies normalize their data and reported that until now, no single method works for everything. We suggest that for panels, using pairwise normalization is a good solution to make results more consistent.
Comment 2:
Abbreviation should arrange in alphabetical order.
Response 2:
Thank you for pointing this out. We fixed it and arranged them in alphabetical order.
Comment 3:
How to get the Figure 2 results?
Response 3:
The results in Figure 2 were obtained by first calculating a panel score (AUC + Sensitivity/100 + Specificity/100) for each study. For every miRNA, we then computed its mean panel score by averaging the scores of all panels that included it. For example, miR-205 appeared in the panels by Yang et al. (2019; score=2.7429) and Leng et al. (2017; score=2.837), giving it a mean score of (2.7429+2.837)/2 ⁓ 2.79. Finally, we ranked all miRNAs by their mean scores and selected the top 30 for visualization in Figure 2.
Moreover, the information on mode of panel score calculation was added to the last section in the article before the figure.
Comment 4:
So many articles about miRNA and cancer were withdraw, so articles related with miRNA was untrustworthy, how do you evaluate the authenticity of your selected literature?
Response 4:
Thank you for your Comment. Our process to ensure authenticity of literature included:
- Selecting studies from major databases and prioritizing the ones with robust methods.
- Emphasizing findings that were consistently reported across multiple studies.
- Using tools like Zotero and EndNote, which provide alerts for retracted articles, to verify the status of all references.

Reviewer 3 Report
Comments and Suggestions for Authors
We appreciate your careful and organized review on blood-based miRNA panels for the early detection of NSCLC. The manuscript presents an excellent overview and the tables and figures are clear and informative for the reader.
Here are a few ways in which the work can be strengthened:
Explain what the novelty of the review is. Certain background sections condense well-known ideas; better stressing what else this review brings to the table (e.g., a ranking of miRNAs and a comparative analysis of signal normalization strategies) would make its contribution more evident.
Explain how AUC, specificity, and sensitivity were combined in the methodology that produced the ranking in Figure 2.
Elaboration on standardization hurdles (e.g., plasma vs. serum) and whether it is realistic to be able to identify stable reference miRNAs across all contexts.
Finally, it is worth expanding the discussion to address the challenges and barriers to clinical translation and implementation of miRNA-based panels in the real-world setting.
This was an overall good review, and we hope that other points raised will make it even clearer, more robust and more practically relevant.
Author Response
Comment 1:
Explain what the novelty of the review is. Certain background sections condense well-known ideas; better stressing what else this review brings to the table (e.g., a ranking of miRNAs and a comparative analysis of signal normalization strategies) would make its contribution more evident.
Thank you for your comment. This review consolidates information from numerous studies to provide a comprehensive overview of miRNA involvement in early-stage NSCLC carcinogenesis. We mapped the key genes targeted by miRNAs at early stages, illustrating their connections and contributions to tumor development. Furthermore, we examined recent multi-miRNA diagnostic panels, outlined significant criteria for selecting miRNAs in future panels, and performed a comparative analysis of the chosen panels. Based on diagnostic metrics, we identified which miRNAs demonstrated the most consistent performance for early NSCLC detection. Additionally, we surveyed the normalization methods employed across previous validation studies, highlighting the current absence of a universal standard which was later discussed in the limitations and conclusions. Ultimately, we are confident that this review can serve as a foundational reference to guide more focused research on multi-miRNA panels for the early detection of NSCLC.
Comment 2:
Explain how AUC, specificity, and sensitivity were combined in the methodology that produced the ranking in Figure 2.
Response 2:
The results in Figure 2 were obtained by first calculating a panel score (AUC + Sensitivity/100 + Specificity/100) for each study. For every miRNA, we then computed its mean panel score by averaging the scores of all panels that included it. For example, miR-205 appeared in the panels by Yang et al. (2019; score=2.7429) and Leng et al. (2017; score=2.837), giving it a mean score of (2.7429+2.837)/2 ⁓ 2.79. Finally, we ranked all miRNAs by their mean scores and selected the top 30 for visualization in Figure 2. The information on the mode of panel score calculation was added to the last section in the article before the figure.
Comment 3:
Elaboration on standardization hurdles (e.g., plasma vs. serum) and whether it is realistic to be able to identify stable reference miRNAs across all contexts.
Response 3:
A central challenge in working with miRNA biomarkers is the lack of methodological standardization. A key pre-analytical hurdle is the difference in miRNA concentration between plasma and serum; the clotting process in serum releases additional miRNAs from platelets, creating systematically different profiles that make direct comparison between studies using different matrices unreliable. Furthermore, identifying a single, stable reference miRNA for normalization across all diseases, sample types, and platforms appears unrealistic, as the expression of commonly used candidates (e.g., miR-16, RNU6B) is itself context-dependent and can be altered by various factors mentioned in the article, such as age, sex, and pathology. Therefore, achieving reproducibility requires standardized sample protocols and a shift towards context-specific validation of normalization strategies, rather than the pursuit of a universal standard. We also suggested that the use of pairwise normalization could be a suitable approach for standardizing multiple miRNAs within the same panel.
However, we consider normalization method study/description requires separate detailed consideration and we have sent the manuscript entitled: Comparison of Normalization Methods for Selecting a Non-Small Cell Lung Cancer Marker Panel from Circulating miRNA RT-qPCR Data to IJMS (we are currently answering reviewers' comments.)
Comment 4:
Finally, it is worth expanding the discussion to address the challenges and barriers to clinical translation and implementation of miRNA-based panels in the real-world setting.
Response 4:
Thank you for your valuable suggestion. Therefore, we decided to write this paragraph at the last section of the article by focusing on the clinical translation.
The added text is as following:
“Despite the promising diagnostic performance of multi-miRNA panels in research studies, their translation into clinical diagnostics faces significant challenges with no panel currently holding FDA or CE approval. This can be due to lack of standardization in pre-analytical and analytical method including sample collection, processing and data normalization, which impairs reproducibility and comparability across studies [156,157]. As mentioned earlier in the article, many candidate panels lack robust validation in large-scale, prospective, and ethnically diverse multicenter cohorts, which is essential to prove generalizability and clinical utility. Finally, the technical complexity and high cost of implementing advanced discovery platforms in routine clinical laboratories present a significant practical challenge [158]. To overcome these challenges, it is recommended to shift the focus from individual miRNA biomarkers to validation of multi-miRNA signatures within prospective, multi-center studies. This required conducting prospective, blinded validation studies across multiple centers and diverse populations using standardized methodologies. Such an approach represents a critical first step toward translating these panels into clinical diagnostics.”

Reviewer 4 Report
Comments and Suggestions for Authors
This is an excellent review focused on the potential of using specific blood miRNAs for early detection of non-small lung cancer (NSLC). This simple and inexpensive approach to NSLC detection is very promising. The authors analyze a vast bulk of literature. A detailed scheme presents known targets of NSLC-related miRNA. The tables are quite informative. The discussion is intelligent and inquisitive. The review will be useful for a wide audience of oncologists and cell/molecular biologists.
Author Response
Comment 1:
This is an excellent review focused on the potential of using specific blood miRNAs for early detection of non-small lung cancer (NSLC). This simple and inexpensive approach to NSLC detection is very promising. The authors analyze a vast bulk of literature. A detailed scheme presents known targets of NSLC-related miRNA. The tables are quite informative. The discussion is intelligent and inquisitive. The review will be useful for a wide audience of oncologists and cell/molecular biologists.
Response 1: Thank you very much for your valuable opinion. We hope that this article will significantly help the researchers when considering new panels or when by translating the existed panels into clinical diagnostics.

Round 2
Reviewer 1 Report
Comments and Suggestions for Authors
The authors have addressed my comments. Thank you
Author Response
Comments 1: The authors have addressed my comments. Thank you
Response 1: Thank you for your significant comments.

Reviewer 2 Report
Comments and Suggestions for Authors
The authors not answered the questions point-to-point.
Author Response
Comment: The authors not answered the questions point-to-point.
Response: Please find below our responses to each of your previous comments. We believe that they will address the points you have raised.
Comment 1: 1. “This research focused on Blood-based miRNA Panels for Timely Detection of Non-Small Cell Lung Cancer: From Biomarker Discovery to Clinical Translation, after check in pubmed, not so many references about this topic(PMID: 29805626 ), this was mean this manuscript was with some Innovation.”
Response 1: Thank you for pointing this out. Starting from 2007 a growing interest to the study of miRNA in NSCLC was observes: more than 400 publications per year 2018-2022 are presented in PubMed for the request “miRNA NSCLC”. Of course, a lot of data were obtained and discussed. Despite of this interest, we agree that some aspects of this topic were not been discussed in depth in previous literature. Our review article highlights several key aspects of the role of miRNAs in diagnosing NSCLC, while also acknowledging that their therapeutic potential remains an important area for exploration. Furthermore, in Response 4, we have addressed the main differences between the selected review article you mentioned (PMID: 29805626, doi: 10.3892/ol.2018.8469) and our manuscript. We believe our review provides a more comprehensive view on application of miRNA for in time NSCLC diagnostics. We are confident it will be of significant interest to readers and we hope that it will serve as a starting point for future research, particularly in developing miRNA panels and identifying better normalization methods.
Comment 2: 2. “Whole manuscript contained so much data and contents, but I think some places can be more perfect.”
Response 2: Thank you for your valuable comment. Based on your feedback and that of the other reviewers, we have revised several sentences and sections of the manuscript to enhance clarity. In particular, the language and the presentation of the conclusions and limitations in the final section have been improved. All changes made to this article after the first round of revisions were highlighted in yellow.
Comment 3: 3. “Early diagnosis is very important for the prognosis of patients with lung cancer. This review summarizes the clinical significance of blood miRNA for early diagnosis of lung cancer.”
Response 3: Thank you again. Indeed, this article provides a comprehensive overview for researchers by presenting the most common validated miRNAs in early-stage NSCLC, summarizing their main pathways and gene targets, and highlighting the importance of miRNAs as non-invasive diagnostic biomarkers compared to traditional methods like Low-Dose Computed Tomography (LD-CT).
Comment 4: 4. “Please clarify the focus and importance of this article compared with the published review “Clinical significance of blood-based miRNAs as biomarkers of non-small cell lung cancer. Oncol Lett. 2018 Jun;15(6):8915-8925. doi: 10.3892/ol.2018.8469.””
Response 4: Thank you for raising this important point. Our review provides a distinct and updated contribution to the field, complementing and extending the valuable work of Li et al. (2018) doi: 10.3892/ol.2018.8469 in Oncology Letters. The key differentiating and novel aspects of our manuscript are: 1. Broadened biomarker scope: While the cited review focused specifically on exosomal miRNAs, our work provides a comprehensive review on cell-free miRNAs (cf-miRNAs) in circulation. Thus, we look on not only exosomes as a source of miRNA originated from tumor cells but miRNA circulated out of exosomes and other membrane covered vesicles as well as in total pool of blood vesicles (including exosomes). 2. Our review provides an updated and comprehensive synthesis by incorporating studies published over the subsequent five years. This allows us to evaluate a substantially larger and more current body of evidence, capturing significant advances since the previous review. Specifically, the 2018 review (with 77 total references) contained only a few studies from 2016-2017. Furthermore, its coverage of early lung cancer diagnostics was limited; for example, the topic was addressed in a general section of 19 lines with 6 references, and the discussion of exosomes comprised 13 lines with 3 references, primarily being descriptive. We recognize that this reflects the scarcity of published data on this topic available in 2018. Our work builds upon this foundation by integrating the wealth of research produced since then. 3. Panel evaluation and miRNA ranking: We examined specific diagnostic panels, explained why they were chosen, and used their results to rank which miRNAs work best. This helps select the most reliable miRNAs for future tests. 4. Normalization strategy analysis was discussed so far as it is drastically concerned with reliable data analysis. We reviewed the different ways studies normalize their data and reported that until now, no single method works for everything. We suggest that for panels, using pairwise normalization is a good solution to make results more consistent.
Comment 5: 5. “Abbreviation should arrange in alphabetical order.”
Response 5: Thank you for pointing this out, we have arranged the abbreviations in alphabetical order as you suggested.
Comment 6: 6. “How to get the Figure 2 results?”
Response 6: Thank you and we agree that it needed more clarification, therefore, we added more details to the last section in the article before the figure. The explain was as following: The results in Figure 2 were obtained by first calculating a panel score (AUC + Sensitivity/100 + Specificity/100) for each study. For every miRNA, we then computed its mean panel score by averaging the scores of all panels that included it. For example, miR-205 appeared in the panels by Yang et al. (2019; score=2.7429) and Leng et al. (2017; score=2.837), giving it a mean score of (2.7429+2.837)/2 ⁓ 2.79. Finally, we ranked all miRNAs by their mean scores and selected the top 30 for visualization in Figure 2. The addition was made to the lines (354-358, page 13) and lines (453-456, page 16)
Comment 7: “7. So many articles about miRNA and cancer were withdraw, so articles related with miRNA was untrustworthy, how do you evaluate the authenticity of your selected literature?”
Response 7: We agree about the importance of this idea and thank you for mentioning this point. However, as we mentioned before, our process to ensure authenticity of literature included selecting studies from major databases and prioritizing the ones with robust methods which are frequently cited across several studies. Moreover, emphasizing findings that were consistently reported across multiple studies. Eventually, during the time of writing the manuscript and searching through databases, we used tools like “Zotero” and “EndNote”, which provide alerts for retracted articles, to verify the status of all references.
